# Antioxidant, Anti-tyrosinase, Anti-α-amylase, and Cytotoxic Potentials of the Invasive Weed *Andropogon virginicus*

**DOI:** 10.3390/plants10010069

**Published:** 2020-12-31

**Authors:** La Hoang Anh, Nguyen Van Quan, Vu Quang Lam, Yu Iuchi, Akiyoshi Takami, Rolf Teschke, Tran Dang Xuan

**Affiliations:** 1Transdisciplinary Science and Engineering Program, Graduate School of Advanced Science and Engineering, Hiroshima University, Hiroshima 739-8529, Japan; hoanganh6920@gmail.com (L.H.A.); nvquan@hiroshima-u.ac.jp (N.V.Q.); iuchiyu0311@gmail.com (Y.I.); 2Division of Hematology, Department of Internal Medicine, Aichi Medical University School of Medicine, Nagakute 480-1195, Japan; quanglamvu1991@gmail.com (V.Q.L.); takami.akiyoshi.490@mail.aichi-med-u.ac.jp (A.T.); 3Department of Internal Medicine II, Division of Gastroenterology and Hepatology, Klinikum Hanau, Teaching Hospital of the Medical Faculty, Goethe University Frankfurt/Main, 63450 Hanau, Germany; rolf.teschke@gmx.de

**Keywords:** *Andropogon virginicus*, antioxidants, α-amylase inhibitor, tyrosinase inhibitor, anti-chronic myeloid leukemia, anti-diabetes, anti-skin aging, high-performance liquid chromatography-electrospray ionization-tandem mass spectrometry, gas chromatography-mass spectrometry

## Abstract

*Andropogon virginicus* is an invasive weed that seriously threatens agricultural production and economics worldwide. In this research, dried aerial parts of *A. virginicus* were extracted, applying Soxhlet and liquid-liquid phase methods to acquire the total crude (T-Anvi), hexane (H-Anvi), ethyl acetate (E-Anvi), butanol (B-Anvi), and water (W-Anvi) extracts, respectively. In which, T-Anvi contains the highest total phenolic and flavonoid contents (24.80 mg gallic acid and 37.40 mg rutin equivalents per g dry weight, respectively). Via anti-radical (ABTS and DPPH), and reducing power assays, E-Anvi exhibits the most potent activities (IC_50_ = 13.96, 43.59 and 124.11 µg/mL, respectively), stronger than butylated hydroxytoluene (BHT), a standard antioxidant, while the lipid peroxidation inhibitory effect of E-Anvi (LPI = 90.85% at the concentration of 500 µg/mL) is close to BHT. E-Anvi shows the most substantial inhibition (IC_50_ = 2.58 mg/mL) on tyrosinase. Notably, α-amylase is significantly suppressed by H-Anvi (IC_50_ = 0.72 mg/mL), over twice stronger than the positive control, palmitic acid. In the cytotoxic assay, E-Anvi is the strongest extract inhibiting K562 cells (IC_50_ = 112.01 µg/mL). Meanwhile, T-Anvi shows the highest prevention on Meg-01 expansion (IC_50_ = 91.40 µg/mL). Dominant compounds detected in E-Anvi by high-performance liquid chromatography-electrospray ionization-tandem mass spectrometry (HPLC-ESI-MS/MS) are identified as flavonoids. However, among four major compounds identified in H-Anvi by gas chromatography-mass spectrometry (GC-MS), palmitic acid and phytol are the most abundant compounds with peak areas of 27.97% and 16.42%, respectively. In essence, this is the first report describing that *A. virginicus* is a potential natural source of antioxidants, tyrosinase and α-amylase inhibitors, and anti-chronic myeloid leukemia (CML) agents which may be useful in future therapeutics as promising alternative medicines.

## 1. Introduction

*Andropogon virginicus* is an invasive weed that is widely distributed in the world. The weed is described as an herbaceous, perennial, C4 plant. *A. virginicus* grows densely and up to 210 cm high [1]. Though this species has been sold in some regions such as Europe, it has been generally considered as not having economic value [2]. Moreover, the strong invasiveness of *A. virginicus* is threatening biodiversity [3,4]. In particular, this weed is problematically competing with endangered plant species like *Tetramolopium remyi*, *Santalum freycinetianum* var. *lanaiense*, and *Schiedea nuttallii* [2,5]. Due its dense and robust growth, *A. virginicus* has also a negative impact on horticultural systems. For instance, it degrades the farmlands of Charmhaven apple in Australia [6]. In addition, *A. virginicus* is dealing a blow economically to the forage and timber production in the southeastern USA because of high competition [2]. In soil, the weed also affects the cycle of nitrogen and water content leading to a decrease in the abundance of native species [7]. Considering that *A. virginicus* is hardly to be controlled and causes numerous serious problems, this invasive weed has received little attention from scientists. Hitherto, no effective strategy has been proposed to control the invasiveness or obtain the benefits of *A. virginicus* for human purposes. Therefore, an effective management of this natural resource is urgently needed, considering the potential of many other plants for therapeutic use, an attractive field of current research. Numerous plant species with antioxidant, anti-inflammatory, antibacterial, antiviral, anti-skin aging, and anticancer properties have been globally known [8]. However, the biological activities related to antioxidant, anti-tyrosinase, anti-α-amylase, and cytotoxic properties of the invasive weed *A. virginicus* have not been studied in detail.

In humans, oxidative stress is tightly linked to inflammation, claimed as a central physiological process in the pathogenesis of numerous chronic diseases including diabetes, aging, and cancer [9]. Concretely, the inflammatory process can exacerbate oxidative stress and vice versa [9,10,11,12,13,14,15]. In fact, many experimental data have indicated the existence and contribution of oxidative stress in several chronic diseases [13,14,15,16,17,18,19,20,21,22]. Therefore, the screening of sample’s antioxidant property is an integral part of our study. Of the serious chronic diseases, diabetes is a complicated disorder with different types requiring varied treatment methods. The most common diabetes is type 2, which causes the elevation of bloodstream sugar level. A potential approach to mitigate type 2 diabetes’ effects can be mentioned to inhibit important enzymes involving in glucose formation, in which α-amylase is a crucial enzyme acting on starch [23]. Additionally, a strong correlation between diabetes and skin manifestations was reported via clinical tests and practical trials [24,25]. Among skin problems, hyperpigmentation resulting in freckles is attributed to the over-formation of melanin through an abnormal tyrosinase activity at both normal melanocytes and malignant melanoma cells [25]. For that reason, the simultaneous inhibition on α-amylase and tyrosinase can be an effective solution to avoid the symptom of type 2 diabetes and skin hyperpigmentation. In cancer cases, chronic myeloid leukemia (CML) is a severe disorder, caused by the unregular proliferation of myeloid cells in the bone marrow. K562 is a typical CML cell line relating to an extraordinary increase in the production of white blood cells [26,27,28]. Meg-01 cell line is also considered to be derived from the myeloid origin, which involve in proliferating nonfunctional immature cells, another type of CML [27,29,30]. Therefore, simultaneous inhibition on both K562 and Meg-01 can be a potential solution in preventing the development of CML. Furthermore, the synergic suppression against oxidative stress, tyrosinase and α-amylase enzymes, and K562 and Meg-01 cell lines could be a prospective approach for developing natural products, which can prevent multiple human health problems.

This research was conducted to look for potential phytochemicals and pharmaceutical properties of the invasive weed *A. virginicus* with focus on antioxidant, anti-skin aging, anti-diabetes, and anticancer properties. Relevant phytochemicals from this invasive species were identified using high-performance liquid chromatography-electrospray ionization-tandem mass spectrometry (HPLC-ESI-MS/MS) and gas chromatography-mass spectrometry (GC-MS).

## 2. Results

### 2.1. Yield and Total Phenolic (TPC) and Flavonoid (TFC) Contents of Extracts from A. virginicus 

From 10 g of *A. virginicus* dried samples, five extracts were obtained including total (T-Anvi), hexane (H-Anvi), ethyl acetate (E-Anvi), butanol (B-Anvi), and water (W-Anvi) extracts. The yield of T-Anvi is 0.98 g (9.8%), followed by E-Anvi (0.32 g, 3.2%), H-Anvi (0.16 g, 1.6%), B-Anvi (0.12 g, 1.2%), and W-Anvi (0.11 g, 1.1%) (Table 1).

The TPC results of extracts from *A. virginicus* range from 0.49 to 25.34 mg GAE/g DW. In which, E-Anvi and T-Anvi show the highest TPC of 25.34 and 24.80 mg GAE/g DW, respectively, which are significantly higher than B-Anvi, H-Anvi, and W-Anvi with TPC of 3.26, 1.26, and 0.49 mg GAE/g DW, respectively. Similarly, the highest TFC is found in T-Anvi (37.40 mg RE/g DW). The following fractional extracts including E-Anvi, H-Anvi, B-Anvi, and W-Anvi account for 68.0%, 16.8%, 13.1%, and 0.8%, respectively, in comparison with TFC of T-Anvi (Table 1). 

### 2.2. Antioxidant Activity of Extracts from A. virginicus

The antioxidant activity of extracts from *A. virginicus* is determined via ABTS, DPPH, β-carotene bleaching, and reducing power assays (Table 2).

In the ABTS assay, the results in Table 2 show that almost samples reveal strong inhibition on ABTS cations, except for H-Anvi and W-Anvi with IC_50_ values of 323.88 and 586.31 µg/mL, respectively. The samples including T-Anvi and B-Anvi show IC_50_ values of 77.71 and 86.73 µg/mL, respectively, which are close to the positive control, BHT, a well-known antioxidant agent. Remarkably, E-Anvi (IC_50_ = 43.59 µg/mL) shows stronger scavenging ability against ABTS cations than BHT.

In the DPPH assay, all samples have scavenging effects on DPPH. In which, IC_50_ values are ranged from 13.96 to 386.91 µg/mL. H-Anvi and W-Anvi repeat the weakest activity with IC_50_ values of 126.27 and 386.91 µg/mL, respectively. While T-Anvi and B-Anvi display similar activity with IC_50_ values of more than 30.00 µg/mL. Significantly, E-Anvi exhibits the highest scavenging capacity against DPPH with IC_50_ value of 13.96 µg/mL, which is approximately twice stronger than BHT (Table 2).

In the case of β-carotene bleaching model, all samples display strong inhibition on lipid peroxidation, which is close to the standard, BHT (LPI = 94.22%), except for W-Anvi (LPI = 81.65%). The strongest sample is B-Anvi (LPI = 92.05%), followed by E-Anvi (LPI = 90.85%), T-Anvi (LPI = 90.81%), and H-Anvi (LPI = 90.17%) (Figure 1).

In the reducing power assay, by comparing IC_50_ values between samples and controls, we obtain that E-Anvi shows the most substantial power (IC_50_ = 124.11 µg/mL), 1.4-fold stronger than the positive standard, BHT (IC_50_ = 175.52 µg/mL). The following extracts are T-Anvi (IC_50_ = 257.35 µg/mL) and B-Anvi (IC_50_ = 340.62 µg/mL). The weakest samples including H-Anvi and W-Anvi reveal trivial reducing power (Figure 2).

### 2.3. Tyrosinase and α-Amylase Inhibitory Activities of A. virginicus Extracts

In the tyrosinase enzymatic assay, the results show that almost samples have inhibitory effects. Among them, the strongest sample is E-Anvi with IC_50_ value of 2.58 mg/mL. The following samples including T-Anvi, H-Anvi, and B-Anvi have IC_50_ values of 4.57, 6.22, and 9.40 mg/mL, respectively. W-Anvi has no effect on the enzyme (Table 3).

While α-amylase is strongly inhibited by the samples including H-Anvi and T-Anvi with IC_50_ values of 0.72 and 3.48 mg/mL, respectively. Notably, H-Anvi is recorded as over twice stronger than palmitic acid, a well-known α-amylase inhibitor. The extracts including E-Anvi and B-Anvi display insignificant suppression on α-amylase. At the concentration of 10 mg/mL, they exhibit the inhibition percentages of 31.93% and 17.52%, respectively. W-Anvi has no effect on α-amylase (Table 3).

### 2.4. Cytotoxic Activity of Extracts from A. virginicus against K562 and Meg-01 Cell Lines

The dose-response curves of *A. virginicus* extracts for cytotoxic effects against K562 and Meg-01 cell lines are presented in Figure 3. Accordingly, *A. virginicus* extracts show inhibitory activity against cell expansion. 

For K562 cell line, E-Anvi exhibits the strongest inhibition with IC_50_ value of 112.01 µg/mL. T-Anvi (IC_50_ = 247.88 µg/mL) is twice weaker than E-Anvi. The remaining fractional extracts reveal insignificant suppression against K562 cell viability.

In the case of Meg-01 cell line, the extracts of T-Anvi, E-Anvi, and H-Anvi reveal substantial cytotoxicity with IC_50_ values of 91.40, 168.94, and 198.07 µg/mL, respectively. While B-Anvi and W-Anvi have negligible prevention against Meg-01 cell viability (Figure 3).

### 2.5. Correlations between Total Phenolic (TPC) and Flavonoid (TFC) Contents and Biological Properties of A. virginicus Extracts

The Pearson’s correlation coefficients between TPC and TFC show that they have a strong association. Additionally, phenolics and flavonoids might be the main contributors to antioxidant activity via ABTS, DPPH, and reducing power assays. However, these compounds might not be involved in lipid peroxidation and α-amylase inhibitory effects of *A. virginnicus* extracts. In contrast, flavonoids might be a vital factor determining the anti-tyrosinase ability. Significantly, the cytotoxicity on K562 and Meg-01 cell lines might be mainly based on the presence of phenolics, especially flavonoid compounds in extracts from *A. virginicus*. The results show that the extract inhibiting K562 cell viability can feasibly suppress Meg-01 as well. Moreover, via all tested assays, the synergic suppression of extracts from *A. virginicus* against various factors involving in human diseases is confirmed through the strong expressed correlation in value of antioxidant, anti-tyrosinase, and K562, Meg-01 cytotoxic activities. The detailed information is presented in Table 4.

### 2.6. GC-MS Results 

From the results of GC-MS chromatogram (Appendix A), there are four abundant compounds detected in H-Anvi, which are reported in Table 5. Among them, palmitic acid is the most abundant compound with a peak area of 27.97%. Phytol is the second dominant compound with a peak area of 16.42%. Other identified substances can be listed as 8-methyl-1-undecene and γ-sitosterol with respective peak areas of 10.77% and 7.38%. The linear retention index (LRI) and Kovats index (KI) were calculated and compared to the literature.

The quantification result shows that palmitic acid accounts for 0.86 mg per 1 g of *A. virginicus* dry weight. While in 1 g of H-Anvi dried extract, 0.05 mg of palmitic acid can be determined.

### 2.7. HPLC-ESI-MS/MS Results

From the chromatogram (Appendix A) and mass spectrum outcomes (Table 6) of HPLC-ESI-MS/MS, eight tentative compounds are found in E-Anvi. In which, five compounds can be identified as flavonoids including kaempferol-O-galactopyranoside, genistin, quercetin-3-O-β-d-glucopyranoside, kaempferol 3-O-β-d-glucopyranoside, eupatilin. The groups of benzophenones (annulatophenonoside), phenolic glycosides (dihydroferulic acid 4-O-glucuronide), proanthocyanidins (prodelphinidin B6) are also detected.

## 3. Discussion

In recent years, the influence of oxidative stress on numerous chronic diseases has been proposed though rarely established [13,14,15,16,17,18,19,20,21,22]. Thus, a vast number of studies were globally conducted to examine the role of antioxidant agents in several chronic disorders such as cardiovascular diseases, cancer, diabetes, Alzheimer’s disease. However, the clinical tests on the human patient have shown failure [31,32,33]. This unsuccessful result can be explained by selecting only potent antioxidant agents to prevent different targeted diseases [34]. Therefore, in the present study, the simultaneous effects of antioxidant, anti-tyrosinase, anti-α-amylase, and cytotoxic potentials were investigated on *A. virginicus*, an invasive weed that was once thought to be of no use.

Basically, the biological activities of plant samples are depended on their chemical composition. Therefore, the detection and identification of phytocompounds are required in the research on plant bioactivity. In the present study, the total phenolic (TPC) and flavonoid (TFC) contents were firstly screened. Among fractional extracts, ethyl acetate contains the highest TPC and TFC. The most abundant compounds identified in the extract belong to flavonoids and flavonoid glycosides. Theoretically, such compounds express a higher affinity to water fraction or butanol fraction. Therefore, previous studies tended to select the polar solvents and extraction protocols such as hexane, ethyl acetate, butanol, 50% ethanol, and water, which were more effective in flavonoid glycoside extraction, and their solubility in ethyl acetate was rather law [35,36]. In fact, the identification and characterization of phytocompound composition of each plant sample are influenced by various factors such as plant species and extraction methods. The TPC and TFC in ethyl acetate extract were found to be equal or higher than aqueous and butanol extracts in case of several species like *Euphorbia splendida* [37] or *Ziziphus jujube* [38]. On the other hand, this study used the Soxhlet extraction with organic solvents (methanol and ethyl acetate 2:8, *v/v*), which could obtain flavonoid glycoside components. This suggests that the mentioned extraction protocol may be appropriate to extract flavonoid glycosides from *A. virginicus* sample. However, further extraction and isolation methods should be applied to optimize the purification of potential compounds from this species. 

In antioxidants, the potent activities of phenolics and flavonoids such as radical scavenging and metal ion chelation have been widely reported. It can be explained via their specific structure of aromatic ring, and especially the number of hydroxyl (-OH) groups [39,40,41,42]. The correlation among TPC, TFC, and antioxidant ability of *A. virginicus* reinforces the above statement (Table 4). Remarkably, the strongest samples of antioxidant activities include the total crude and ethyl acetate extracts, which are recorded to contain high levels of phenolic compounds, particularly flavonoids (Table 1). Notably, the ethyl acetate extract exhibits stronger radical scavenging capacities via ABTS, and DPPH assays and higher reducing power than BHT, a well-known antioxidant agent (Table 2). Hence, the HPLC-ESI-MS/MS method was applied to analyze the chemical profiles in this extract. Accordingly, the ethyl acetate extract is found rich in flavonoid content, which may determine the potent antioxidant ability of this extract (Table 6).

Besides the antioxidant activity, flavonoids in *A. virginicus* may be a main contributor to anti-tyrosinase ability of the total crude and ethyl acetate extracts in this study (Table 3, Table 4 and Table 6). Previously, flavonoids were recorded having inhibitory effects against tyrosinase through in silico and in vitro screenings [43]. However, unlike the antioxidant capacities, the enzymatic reactions are much more complicated and difficult to predict, which are influenced by various features in chemical structures [44]. For example, the rings A and C, (-OH) substituent at the position of 7, and the ring B linked to (-OH) groups at the position of para or para and meta in the flavonoid structures lead to the competitive inhibition against tyrosinase [43]. In the case of anti-α-amylase activity, the inhibition may depend on different factors such as the position of (-OH) substituents, methoxy groups, lactone rings, or the interaction between compounds in the mixture [44]. Via the results of α-amylase inhibitory assay (Table 3), the hexane extract from *A. virginicus* shows the strongest suppression. Thus, the GC-MS analysis was performed to find out the major compounds involving in the prevention against α-amylase in this extract. Among the most dominant compounds, palmitic acid shows the highest peak area and accounts for 0.86 and 0.05 mg per 1 g of *A. virginicus* dry weight and hexane dried extract, respectively (Table 5), whilst other compounds in Table 5 were neither successfully isolated nor be purchased, thus need further clarification. Notably, palmitic acid is an acknowledged α-amylase inhibitor [45,46,47], thus it was used as a positive control in this study. Remarkably, the hexane extract reveals over twice stronger inhibitory effect than palmitic acid by comparing their IC_50_ values. Because palmitic acid is a major compound (Table 5) in the plant, it is suggested that this compound may play an important role in the biological potential of the invasive weed, beside the α-amylase inhibitory effect as shown in this study. The hexane extract’s activity may be strengthened due to the interaction between four dominant compounds consisting of palmitic acid, γ-sitosterol, phytol, and 8-methyl-1-undecene (Table 5). In which, γ-sitosterol is also a high potential anti-diabetes substance, which has been affirmed through in vitro and in vivo tests on rats in recent studies [48]. Therefore, this compound may play an influential role in the anti-α-amylase ability of the hexane extract.

The detected flavonoids in the *A. virginicus*’s total crude and ethyl acetate extracts may be also the primary agents for the cytotoxic activities against K562 and Meg-01 cell lines. According to the previous studies including laboratory bioassays, epidemiological investigations, and clinical trials, flavonoids have shown important chemotherapy roles for cancer. The anticancer property of flavonoids has been comprehensively explained through various biological pathways such as antioxidants, enzymatic activities, genetic factors, and cell-related mechanisms, or synergic effects of these mentioned factors [49]. Remarkably, the antioxidant activity of flavonoids is closely associated with anticancer regulation [50]. Meanwhile, phytoflavonoids are not harmful to normal myeloid, peripheral blood, and epithelial cell lines [51]. In our research, we also find a strong correlation between total phenolics and flavonoids, antioxidant and cytotoxic capacities in the ethyl acetate extract (Table 4). On the other hand, the hexane extract’s cytotoxic activity against Meg-01 cell line may be due to the presence of phytol and γ-sitosterol or the combined influence of the found compounds in this extract (Figure 3 and Table 5). Particularly, phytol is a popular anticancer agent, which has been mentioned in numerous recent studies. The inhibition of this compound on different types of cancer cells and relevant modes of action have been globally published [52,53,54,55,56,57]. Whilst γ-sitosterol has been also known to be an antitumor agent, its specific mechanism has been reported in other studies [58,59].

In addition to finding out potential therapeutic properties and phytocompounds, the specific direction of developing natural-based drugs is essentially considered to come up with the most effective approach [60]. For example, clinical tests through randomized controlled trials are required before the products are registered as drugs. One of the critical factors is that natural products must satisfy the safety requirements. In fact, some of them have shown hepatotoxicity with high application dose in clinical trials on humans [60,61,62]. Therefore, although numerous plant products have proven their potentials for therapeutic uses, very few of them have been officially registered and widely applied. An effective approach can be addressed such as investigating the biological properties of folk medicines and traditional food stuffs. Ginger shell (*Alpinia zerumbet*) can be an example, which is considered as a significant contributor to human longevity in Japanese Okinawa [63,64]. This plant species is popularly used with even no clinical data available. However, the overuse of these natural plants also leads to the loss of biodiversity [60]. Therefore, changing the conventional objects to other available resources, especially problematic invasive species, is a prospective direction. The exploitation of invasive weed for medicinal purposes can simultaneously solve the problems in biodiversity conservation and sustainable development.

In general, the invasive weed *A. virginicus* shows the potential for antioxidant, anti-diabetes, anti-skin aging, and anti-CML properties via in vitro approaches. Therefore, in vivo tests and clinical trials should be further performed to develop novel natural-based drugs. Additionally, the screening results by GC-MS (Table 5) and HPLC-ESI-MS/MS (Table 6) indicate the most dominant compounds including palmitic acid, γ-sitosterol, phytol, and 8-methyl-1-undecene, flavonoids, and flavonoid glycosides, which might be the main contributors to the biological activities of this plant. Based on that, the research on confirmation of their role in biological and pharmaceutical properties of *A. virginicus* should be conducted. Furthermore, the quantification of bioactive compounds in mixtures and extracts showing the potent activity should be also carried out. Palmitic acid has been reported as an inhibitor for anti-diabetes [45,46,47], phytol and γ-sitosterol are popularly known as anticancer agents [52,53,54,55,56,57,58,59]. While flavonoids and flavonoid glycosides can be used as antioxidant agents and applied in treatment for numerous diseases [39,40,41,42,43,49]. The quantification of dominant and specific bioactive compounds should use advantage analytical instruments such as LC-MS and LC-ESI-MS/MS rather than the estimation of total flavonoids and phenols and quantification by only GC-MS. Future studies focusing on isolation, purification, and quantification of bioactive compounds from *A. virginicus* should be performed to explore the medicinal and pharmaceutical potential of this invasive species. This could help farmers in developing countries to improve their income from the utilization of this harmful weed as a source of antioxidants, anti-skin aging, anti-diabetes, and anticancer.

## 4. Materials and Methods

### 4.1. Materials 

#### 4.1.1. Chemicals and Cell Lines

Extraction solvents including methanol, hexane, ethyl acetate, and butanol were purchased from Junsei Chemical Co., Ltd., Tokyo, Japan. Standards including gallic acid, rutin, butylated hydroxytoluene (BHT), kojic acid were purchased from Kanto Chemical Co., Inc., Tokyo, Japan. Palmitic acid was obtained from FUJIFILM Wako Pure Chemical Corporation.

Chemicals including sodium hypochlorite (NaClO), Folin-Ciocalteu’s reagent, sodium carbonate (Na_2_CO_3_), aluminum chloride (AlCl_3_), 2,2′-azinobis-(3-ethylbenzothiazoline-6-sulfonic acid) (ABTS), potassium persulfate (K_2_S_2_O_8_), 2,2-diphenyl-1-picrylhydrazyl (DPPH), sodium acetate (CH_3_COONa), β-carotene, chloroform, linoleic acid, tween 40, dipotassium phosphate (K_2_HPO_4_), monopotassium phosphate (KH_2_PO_4_), potassium ferricyanide (K_3_[Fe(CN)_6_]), trichloro acetic acid (CCl_3_COOH), ferric chloride (FeCl_3_), dimethyl sulfoxide (DMSO), monosodium phosphate (NaH_2_PO_4_), sodium chloride (NaCl), hydrochloric acid (HCl) were obtained from Kanto Chemical Co., Inc., Tokyo, Japan. While tyrosinase, L-tyrosine, α-amylase, starch, iodine, 3-(4,5-dimethylthiazolyl)2,5-diphenyl-tetrazolium bromide (MTT), Iscove’s Modified Dulbecco’s Medium (IMDM), and cell lysis buffer were purchased from Sigma-Aldrich, St. Louis, MO, USA.

K562 and Meg-01 cell lines were obtained from American Type Culture Collection (ATCC), Virginia, United States.

#### 4.1.2. Plant Materials

The aerial plant parts of *A. virginicus* were collected in fields affiliated to Higashi-Hiroshima Campus, Hiroshima University, Hiroshima, Japan in October 2019. The species’ identification was primarily based on the publication of European and Mediterranean Plant Protection Organization [2], available database of Missouri Botanical Garden, United States [65], the National Institute for Environmental Studies, Japan [66], and the book entitled: “Invasive Weed” published by Japan Livestock Technology Association [67]. The specimen with voucher number of Anvi-J2019 was preserved at the Laboratory of Plant Physiology and Biochemistry, Graduate School of Advanced Science and Engineering, Hiroshima University, Japan.

### 4.2. Sample Preparation and Extraction

Impurities were removed from *A. virginicus* aerial plant parts by water and 0.5% NaClO before being dried by an oven at 40 °C for 1 week. Fine powder was made by grinding the dried samples. Subsequently, the obtained powder (10 g) was subjected to a Soxhlet extractor consisting of extraction chamber and condenser (Climbing Co., Ltd., Fukuoka, Japan) connecting to a mantle heater (Daika Denki Co., Ltd., Osaka, Japan). An aliquot (400 mL) of methanol and ethyl acetate (2:8, *v/v*) mixture was the extraction solvent. After 5 h, the obtained extract was filtrated and concentrated by removing solvent with a rotary evaporator (Nihon Buchi K.K., Tokyo, Japan) to yield the crude extract (T-Anvi). After that, the dried T-Anvi was mixed with distilled water (50 mL) and subjected to liquid-liquid phase (1:1, *v/v*) extraction with hexane, followed by ethyl acetate, and butanol, respectively. With each extraction solvent, the aqueous phase was returned to the separatory funnel, and the entire process was repeated to collect the maximum amount of target compounds. Finally, the collected extracts were filtered and evaporated at 50 °C under vacuum to acquire the dried hexane (H-Anvi), ethyl acetate (E-Anvi), and butanol (B-Anvi) extracts, respectively. The remaining water residue was also concentrated to attain the water extract (W-Anvi).

### 4.3. Total Phenolic (TPC) and Flavonoid (TFC) Contents

The Folin-Ciocalteu method was applied to determine the TPC of extracts from *A. virginicus* [68]. Briefly, the mixture including 20 µL of sample (in methanol), 100 µL of 10% Folin-Ciocalteu solution and 80 µL of 7.5% Na_2_CO_3_ was incubated at room temperature for 30 min. The result was recorded at 765 nm and expressed as milligrams of gallic acid equivalent per one gram of sample dry weight (mg GAE/g DW).

The aluminum chloride colorimetric method represented by Tuyen et al. [69] was applied to evaluate the TFC of extracts from *A. virginicus* with a slight alteration. In detail, the incubated reaction of 50 µL of sample (in methanol) and an equal volume of 2% AlCl_3_ was carried out. The result was measured after 15 min at 430 nm and presented as milligrams of rutin equivalent per one gram of sample dry weight (mg RE/g DW).

### 4.4. Antioxidant Activity

#### 4.4.1. ABTS Cation Decolorization Assay

The method was presented by Pellegrini et al. [70]. In advance of colorimetric reaction, the ABTS working solution was prepared in the course of incubation for the mixture consisting of 7 mM ABTS and 2.45 mM K_2_S_2_O_8_ (1:1, *v/v*) for 16 h without light at 25 °C. Methanol was then added to achieve the targeted absorbance (0.65–0.75) at 734 nm. The decolorization ability was evaluated by pipetting sample/or control (in methanol) into ABTS working solution (1:9, *v/v*) with a total volume of 200 µL of combination. The reaction was incubated in darkness for 20 min. The inhibition percentage was measured as decreased absorbance over the control at 734 nm.

#### 4.4.2. DPPH Radical Scavenging Assay

Based on the protocol demonstrated by Elzaawely and Tawata [71], the discoloration ability of 80 µL of methanolic sample/or control on 40 µL of DPPH solution (0.2 mg/mL) in addition to 80 µL of acetate buffer (0.1 M, pH = 5.5) was observed. The absorbance was read after 20 min of incubation at 25 °C with an avoidance of direct light. The inhibition was recorded as percentage of decreased absorbance over the control at 517 nm.

The anti-radical (ABTS and DPPH) ability was evaluated following the below formula:Anti-radical ability (%) = (C − S)/C × 100(1)C: Absorbance of reaction with negative control (methanol) after subtracting control’s blank (without radical solution), S: Absorbance of reaction with sample/or positive control (BHT) after subtracting sample’s blank (without radical solution).

Different concentrations of sample/or control were tested to establish a dose-dependent curve (linear equation). Based on the equation, the required concentration of samples/or control for inhibiting 50% of radicals (IC_50_ values) was obtained. Lower IC_50_ value means stronger inhibition.

#### 4.4.3. β-Carotene Bleaching Assay 

The process was performed according to the previous report of Tuyen et al. [69]. Firstly, the combination containing β-carotene (200 μg), chloroform (1 mL), linoleic acid (20 μL), and tween 40 (200 mg), respectively was prepared. The obtained solution was then concentrated under vacuum at 40 °C by a rotary evaporator. Subsequently, fifty milliliters of oxygenated water were slowly mixed and shaken to generate a stable emulsion. The lipid peroxidation inhibition (LPI) was evaluated by adding methanolic sample/or control (500 µg/mL) to the working emulsion (1:8, *v/v*) in a total of 225 μL of reaction solution. Similar to anti-radical assay, methanol and BHT were used as controls. The absorbance readings were recorded every 15 min for a total of 13 times at 492 nm. The lipid peroxidation inhibition (LPI) was measured as follows: LPI (%) = A_180_/A_0_ × 100(2)A_180_: absorbance of reaction at the 180^th^ min, A_0_: absorbance of reaction at the beginning time point.

#### 4.4.4. Reducing Power Assay

We followed the same method applied in the study of Quan et al. [44]. For preparation, the solutions including phosphate buffer (0.2 M, pH = 6.6); potassium ferricyanide (1 g/100 mL); trichloro acetic acid (10 g/100 mL); ferric chloride (0.01 g/10 mL) were prepared in distilled water. The first step was to incubate the mixture of methanolic sample/or control (0.1 mL), phosphate buffer (0.25 mL), and potassium ferricyanide solution (0.25 mL) for 30 min at 50 °C. After that, trichloro acetic acid solution (0.25 mL) was added and centrifuged for 10 min at 4000 rpm. Subsequently, the supernatant (100 μL) was mixed with 100 µL of distilled water, followed by 20 µL of ferric chloride solution. The absorbance of final solution was read at 700 nm. Methanol and BHT were used as negative and positive controls, respectively. Five concentrations (10, 62.5, 125, 250, 500 µg/mL) of sample/or control were examined to build the dose-responding curves. From the curve, IC_50_ values were calculated to compare between samples. In which, IC_50_ value was the required concentration to reach the absorbance of 0.5.

### 4.5. Enzymatic Inhibitory Assays 

#### 4.5.1. Tyrosinase Inhibition

The assay components included potassium phosphate buffer (20 mM, pH = 6.8), tyrosinase (500 units/mL in buffer), L-tyrosine substrate (2 mM in distilled water) solutions. Samples/or controls were prepared in DMSO. At the beginning, 20 µL of sample/or control and 20 µL of enzyme solution were homogenized in 120 µL of buffer. The combination was incubated at 25 °C for 5 min. After that, the enzymatic reaction was inaugurated by supplying 50 µL of L-tyrosine solution. After 10 min of incubation at the same condition, the final solution’s absorbance was recorded at 470 nm [72]. DMSO and kojic acid were used as negative control and standard inhibitor, respectively. The inhibitory activity was calculated using the formula as follows:% inhibition = (C − S)/C × 100(3)C: Absorbance of reaction with DMSO after subtracting control’s blank (without enzymatic reaction on substrate), S: Absorbance of reaction with sample/or inhibitor after subtracting sample’s blank (without enzymatic reaction on substrate).

IC_50_ values of samples against tyrosinase were recorded in the same method with anti-radical assay.

#### 4.5.2. α-Amylase Inhibition 

The in vitro α-amylase inhibition assay was determined according to the method reported by Quan et al. [44]. For preparation, phosphate buffer saline (0.2 M, pH = 6.9) and α-amylase (5 units/mL in buffer), starch substrate (0.5% in distilled water), iodine (0.25 mM in distilled water) solutions were included. Samples/or controls were prepared in the buffer. Initially, the incubation of 40 µL of mixture including sample/or control and prepared enzyme (1:1, *v/v*) was conducted at 37 °C for 9 min. The enzymatic reaction on substrate was generated by providing 30 µL of starch working solution. The reaction was continued at 37 °C for 7 min before stopping by 20 µL of HCl (1 M). To detect the remaining amount of substrate in the final solution, one hundred microliters of prepared iodine solution were added. The final reaction mixture was recorded at 565 nm. DMSO (50% in buffer) was negative control and palmitic acid was standard inhibitor. The inhibition percentage was determined as follows:% inhibition = S/C × 100(4)S: Absorbance of reaction with sample/or inhibitor after subtracting an absorbance of the enzymatic reaction without inhibitor, C: Absorbance of reaction without sample/or control and enzyme after subtracting an absorbance of the enzymatic reaction without inhibitors.

Similar to anti-radical and tyrosinase assays, IC_50_ values were applied to compare inhibitory activities among samples.

### 4.6. Cytotoxic Activity against K562 and Meg-01 Cell Lines

The cell growth rate was determined through the cell metabolic activity applying MTT assay. The cell culture media was IMDM containing fetal bovine serum (10%), L-glutamine (5 mM), Penicillin (100 IU/mL), and Streptomycin (100 µg/mL). Samples were prepared in 0.1% DMSO (in culture media). In each plate’s well, the cells (5 × 10^3^) were seeded in culture media (90 µL) and placed in an incubator (Thermo Fisher Scientific, Waltham, United States) for 24 h at 37 °C with CO_2_ 5%. After that, ten microliters of sample were pipetted to each well. After 40 h, ten microliters of MTT solution (5 mg/mL) were supplied. The cells were continuously incubated at 37 °C for 4 h. Subsequently, one hundred microliters of cell lysis buffer (10% SDS in 0.01 M HCl) were added to each well. The cell expansion was observed using an inverted microscope (LabX, Midland, Canada). The absorbance at 595 nm was applied to determine the cell viability in value. The negative control was conducted by adding 10 microliters of culture media instead of sample.
% inhibition = (C − S)/C × 100(5)C: Absorbance of the reaction with negative control, S: Absorbance of reaction with sample.

The dose-response curves and IC_50_ values of samples against K562 and Meg-01 cell lines were established to compare between samples.

### 4.7. Identification of Phytochemical Component 

#### 4.7.1. GC-MS

The phytochemical components of the hexane extract (H-Anvi) were analyzed by GC-MS system (JMS-T100 GCV, JEOL Ltd., Tokyo, Japan). The column (DB-5MS, 30 m × 0.25 mm I.D. × 0.25 µm film thickness) (Agilent Technologies, J&W Scientific Products, Folsom, United States) was connected to the system. The injection was performed by the system equipped with an autosampler. Sample (2 mg/mL) was prepared for measurement. The split ratio of 5:1 was applied for the carrier gas helium. The GC oven’s temperature was scheduled as follows: (1)Initial temperature: 50 °C (without holding),(2)Rushing temperature: 10 °C/min for 30 min and 20 min in maintenance,(3)Injector and detector temperatures: 300 °C and 320 °C, respectively.

The mass was scanned in the range of 29 to 800 amu. The results of chromatogram, mass spectrum (electron ionization), linear retention index (LRI), and Kovats index (KI) of each detected compound were collected. The outcomes were compared to the standard sources comprising the library of JEOL’s GC-MS Mass Center System Version 2.65a, and the online database of National Center for Biotechnology Information, U.S. National Library of Medicine, Bethesda MD, USA (Pubchem—https://pubchem.ncbi.nlm.nih.gov), and the National Institute of Standards and Technology, U.S. Department of Commerce (NIST—https://webbook.nist.gov). Based on the GC-MS results, palmitic acid was detected as the most dominant compound in H-Anvi. Therefore, a calibration curve applying different concentrations of palmitic acid standard (0.1, 0.5, and 1 mg/mL) was established to determine the content of palmitic acid in *A. virginicus*. The result was expressed as milligrams per one gram of sample dry weight (mg/g DW).

#### 4.7.2. HPLC-ESI-MS/MS

The phytochemicals of the ethyl acetate extract (E-Anvi) were identified based on the HPLC-ESI-MS/MS method. In particular, the HPLC-ESI-MS/MS system was equipped with LTQ Orbitrap XL mass spectrometers (Thermo Fisher Scientific, Waltham, United States) and an electrospray ionization (ESI) source. For analysis, sample (2 mg/mL) with a volume of 3.0 μL was injected into the XBridge® Shield RP18 (5 μm, 2.1 × 100 mm) column, which maintained the temperature at 25 °C. The gradient model with 2 solvents was set up as follows: solvent A was 0.1% formic acid in water, solvent B was pure acetonitrile. The gradient program was: 95% A and 5% B over 0–2 min, then changed to 30% A and 70% B over 2–12 min, followed by 0% A and 100% B over 12–22 min. From 22–34 min, the mobile phase was subjected to the initial condition of 95% A and 5% B. The flow rate was 400 μL/min. For mass scanning, a negative FTMS mode with the range from 100 to 700 m/z was applied. The online database (Pubchem, National Center for Biotechnology Information, U.S. National Library of Medicine, Bethesda MD, USA) was used for reference to MS/MS spectra.

### 4.8. Data Analysis

All bioassays were tested with three replications. One-way ANOVA statistical method in Minitab software version 16.2.3 was applied to analyze the raw data. The statistical data including means, standard deviations (SD), and grouping information were obtained and expressed as final outcomes. Various groups indicate significant differences under Turkey’s test at the confidence level of 95%. Pearson’s correlation coefficients between total phenolic and flavonoid contents, antioxidant, anti-tyrosinase, α-amylase, and cytotoxic activities of extracts from *A. virginicus* were evaluated using the same software. 

## 5. Conclusions

This is the first investigation of the biological properties of the invasive weed *A. virginicus* including antioxidants, anti-α-amylase, and anti-tyrosinase abilities, and cytotoxicity against CML cell lines. The results show that the ethyl acetate extract from *A. virginicus* exhibits the highest antioxidant activity via ABTS, DPPH, reducing power assays, and β-carotene bleaching models. In addition, this extract reveals potential anti-tyrosinase capacity and potent cytotoxicity against K562 and Meg-01 cell lines. Meanwhile, the hexane extract displays strong α-amylase and Meg-01 inhibitory effects. The chemical analysis results indicate that *A. virginicus* aerial parts are rich in flavonoids, palmitic acid, phytol, and γ-sitosterol, which may principally play a vital role in biological activities of the respective extracts. The finding suggests that *A. virginicus* is a promising source of antioxidant, anti-diabetic, anti-tyrosinase, and antitumor agents. Further in vivo and clinical tests should be performed to confirm and develop the natural functional products from *A. virginicus* for pharmaceutical purposes.

## Figures and Tables

**Figure 1 plants-10-00069-f001:**
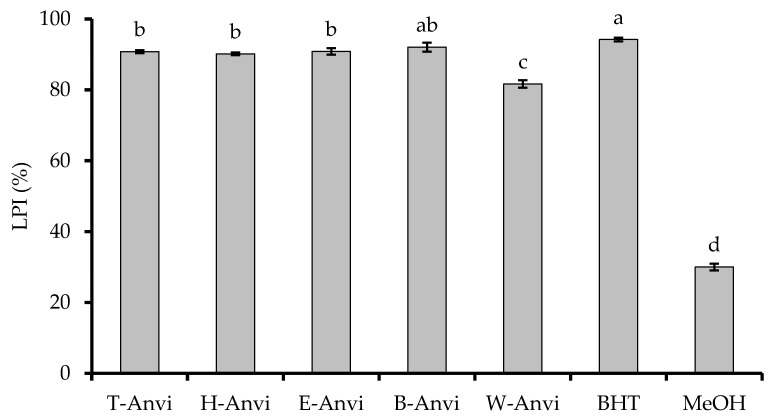
Inhibition on lipid peroxidation of *A. virginicus* extracts. Different letters indicate significant differences at *p* < 0.05. LPI, lipid peroxidation inhibition; T-Anvi, total crude extract; H-Anvi, hexane extract; E-Anvi, ethyl acetate extract; B-Anvi, butanol extract; W-Anvi, water extract; BHT, butylated hydroxytoluene.

**Figure 2 plants-10-00069-f002:**
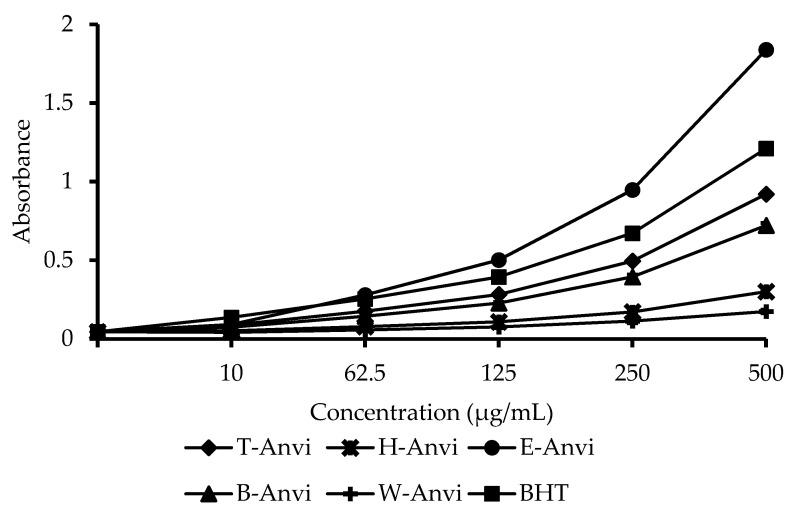
Potassium ferricyanide reducing power of *A. virginicus* extracts. T-Anvi, total crude extract; H-Anvi, hexane extract; E-Anvi, ethyl acetate extract; B-Anvi, butanol extract; W-Anvi, water extract; BHT, butylated hydroxytoluene.

**Figure 3 plants-10-00069-f003:**
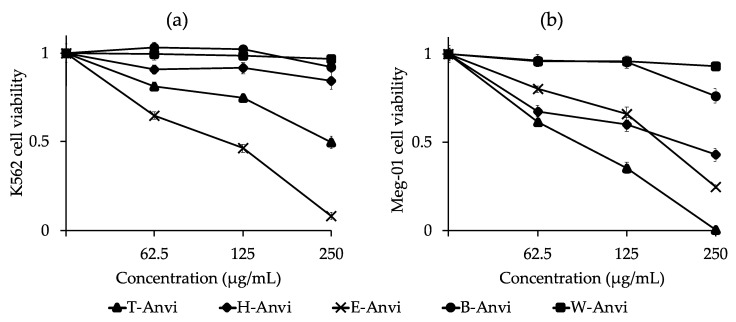
Dose-response curves of *A. virginicus* extracts for cytotoxicity against (**a**) K562 and (**b**) Meg-01 cell lines. T-Anvi, total crude extract; H-Anvi, hexane extract; E-Anvi, ethyl acetate extract; B-Anvi, butanol extract; W-Anvi, water extract.

**Table 1 plants-10-00069-t001:** Extraction yield and total phenolic (TPC) and flavonoid (TFC) contents.

Samples	Code	Extraction Yield (%)	TPC (mg GAE/g DW)	TFC (mg RE/g DW)
Total crude extract	T-Anvi	9.8	24.80 ± 0.51 ^a^	37.40 ± 0.74 ^a^
Hexane extract	H-Anvi	1.6	1.26 ± 0.03 ^c^	6.30 ± 0.13 ^c^
Ethyl acetate extract	E-Anvi	3.2	25.34 ± 0.47 ^a^	25.44 ± 0.45 ^b^
Butanol extract	B-Anvi	1.2	3.26 ± 0.06 ^b^	4.91 ± 0.08 ^d^
Water extract	W-Anvi	1.1	0.49 ± 0.01 ^c^	0.25 ± 0.01 ^e^

Data express means ± standard deviation (SD). Different superscript letters in a column indicate significant differences at *p* < 0.05. TPC, total phenolic content; TFC, total flavonoid content; GAE, gallic acid equivalent; RE, rutin equivalent; DW, dry weight.

**Table 2 plants-10-00069-t002:** Anti-radical activities of *A. virginicus* extracts.

Samples	ABTS Assay IC_50_ (µg/mL)	DPPH Assay IC_50_ (µg/mL)
T-Anvi	77.71 ± 1.85 ^d^	30.54 ± 0.40 ^cd^
H-Anvi	323.88 ± 1.22 ^b^	126.27 ± 4.92 ^b^
E-Anvi	43.59 ± 1.25 ^f^	13.96 ± 0.39 ^e^
B-Anvi	86.73 ± 2.30 ^c^	33.79 ± 0.24 ^c^
W-Anvi	586.31 ± 6.11 ^a^	386.91 ± 8.24 ^a^
BHT	63.51 ± 0.43 ^e^	20.81 ± 0.67 ^de^

Data express means ± standard deviation (SD). Different superscript letters in a column indicate significant differences at *p* < 0.05. DPPH, 2,2-diphenyl-1-picrylhydrazyl; ABTS, 2,2′-azinobis-(3-ethylbenzothiazoline-6-sulfonic acid); T-Anvi, total crude extract; H-Anvi, hexane extract; E-Anvi, ethyl acetate extract; B-Anvi, butanol extract; W-Anvi, water extract; BHT, butylated hydroxytoluene.

**Table 3 plants-10-00069-t003:** Tyrosinase and α-amylase inhibitory activities of *A. virginicus* extracts.

Samples	Tyrosinase Inhibition IC_50_ (mg/mL)	α-Amylase Inhibition IC_50_ (mg/mL)
T-Anvi	4.57 ± 0.05 ^c^	3.48 ± 0.07 ^a^
H-Anvi	6.22 ± 0.08 ^b^	0.72 ± 0.01 ^c^
E-Anvi	2.58 ± 0.13 ^d^	ne
B-Anvi	9.40 ± 0.02 ^a^	ne
W-Anvi	na	na
Kojic acid	0.02 ± 0.00 ^e^	-
Palmitic acid	-	1.57 ± 0.04 ^b^

Data express means ± standard deviation (SD). Different superscript letters in a column indicate significant differences at *p* < 0.05. T-Anvi, total crude extract; H-Anvi, hexane extract; E-Anvi, ethyl acetate extract; B-Anvi, butanol extract; W-Anvi, water extract; na, no activity; ne, negligible effect; -, not determined.

**Table 4 plants-10-00069-t004:** Pearson’s correlation coefficients between total phenolic and flavonoid contents and biological activities of *A. virginicus* extracts.

	TPC	TFC	ABTS	DPPH	β-Ca	RP	Tyro	α-Amy	K562
**TFC**	0.952 *	-	-	-	-	-	-	-	-
**ABTS**	0.833 *	0.674 *	-	-	-	-	-	-	-
**DPPH**	0.813 *	0.633 *	0.993 *	-	-	-	-	-	-
**β-Ca**	0.423	0.466	0.589 *	0.549 *	-	-	-	-	-
**RP**	0.843 *	0.667 *	0.987 *	0.997 *	0.523 *	-	-	-	-
**Tyro**	0.479	0.525 *	0.601 *	0.572 *	0.970 *	0.555 *	-	-	-
**α-Amy**	−0.236	−0.043	−0.439	−0.416	0.217	−0.391	0.302	-	-
**K562**	0.898 *	0.736 *	0.891 *	0.914 *	0.330	0.942 *	0.393	−0.307	-
**Meg-01**	0.787 *	0.913 *	0.404	0.378	0.435	0.424	0.526 *	0.351	0.554 *

*, a significance at *p* < 0.05; TPC, total phenolic content; TFC, total flavonoid content; ABTS, 2,2′-azinobis-(3-ethylbenzothiazoline-6-sulfonic acid) assay; DPPH, 2,2-diphenyl-1-picrylhydrazyl assay; β-Ca, β-carotene bleaching assay; RP, reducing power assay; Tyro, tyrosinase inhibitory assay; α-Amy, α-amylase inhibitory assay; K562, K562 cytotoxic assay; Meg-01, Meg-01 cytotoxic assay.

**Table 5 plants-10-00069-t005:** Dominant phytochemicals in H-Anvi extract from *A. vigirnicus* identified by GC-MS.

No.	Identified Compound	RT (min)	MW	Formula	Classification	Peak Area (%)	LRI	KI	Content (mg/g DW)
1	Phytol	15.80	296	C_20_H_40_O	Diterpenoids	16.42	1835	1835	-
2	8-Methyl-1-undecene	16.64	168	C_12_H_24_	Alkenes	10.77	1916	1917	-
3	Palmitic acid	17.02	256	C_16_H_32_O_2_	Fatty acids	27.97	1955	1955	0.86
4	γ-Sitosterol	28.34	414	C_29_H_50_O	Steroids	7.38	-	-	-

RT, retention time; LRI, linear retention index; KI, Kovats index; DW, dry weight; -, not determined.

**Table 6 plants-10-00069-t006:** Phytocompounds in E-Anvi extract from *A. virginicus* detected by HPLC-ESI-MS/MS.

No. Signals	RT (min)	[M+H]^−^ (m/z)	Chemical Classification	Tentative Identity	Molecular Formula	Exact Mass	Fragmentions (m/z)
1	0.68	377.086	Benzophenones	Annulatophenonoside	C_18_H_18_O_9_	378.3	89.024(100); 119.086(53); 143.036(45); 149.099(28)
2	5.74	371.098	Phenolic glycosides	Dihydroferulic acid 4-O-glucuronide	C_16_H_20_O_10_	372.3	231.158(100); 243.293(23); 225.296(19); 121.105(14)
3	6.08	447.093	Flavonoids	Kaempferol-O-galactopyranoside	C_21_H_20_O_11_	448.4	327.154(100); 357.141(48); 429.155(5); 369.242(4)
4	6.35	431.098	Flavonoids	Genistin	C_21_H_20_O_10_	432.4	311.140(100); 341.117(6); 283.142(2)
5	6.54	463.088	Flavonoids	Quercetin-3-O-β-d-glucopyranoside	C_21_H_20_O_12_	464.4	445.145(100); 343.148(92); 427.169(75); 373.161(48)
6	6.67	447.093	Flavonoids	Kaempferol 3-O-β-d-glucopyranoside	C_21_H_20_O_11_	448.4	285.167(100); 402.158(6); 428.144(5); 374.128(3)
7	8.37	413.088	Proanthocyanidins	Prodelphinidin B6	C_21_H_18_O_9_	414.4	313.135(100); 297.155(73); 369.145(70); 285.154(20)
8	9.71	343.082	Flavonoids	Eupatilin	C_18_H_16_O_7_	344.3	311.153(100)

RT, retention time.

## Data Availability

All data are presented in the article and Appendix A.

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
