# Peer review of "Antioxidant, Anti-tyrosinase, Anti-α-amylase, and Cytotoxic Potentials of the Invasive Weed Andropogon virginicus"

_plants, 2020, doi:10.3390/plants10010069_

Round 1
Reviewer 1 Report
The manuscript presents interesting data of the determination of phenolic and flavonoid content in fractions obtained from aerial parts of the invasive weed, Andropogon virginicus. To determine the biological effect of particular fractions, antiradical activity, anti-tyrosinase and anti-⍺-amylase effect, inhibition of lipid peroxidation, and cytotoxic effects were evaluated. Additionally, the lipid profile in the hexane fraction was determined using GC-MS. The authors utilized an appropriate technique of characterization and analysis, supplying useful information. Overall the work is well written and organized, although some issues should be discussed or corrected. The changes and suggestions are listed below. General remarks: 1) In the presented research, ethyl acetate fraction contained the highest amount of phenolic and flavonoids. The subsequent fractionation with n-butanol, alcohol of twice higher than ethyl ether relative polarity in relation to water, allowed to achieve the lower amounts of phenolic and flavonoids. The compound detected in the analyzed fraction is phenolic and flavonoids with many –OH functional groups in the basic aglycone skeleton. A few are characterized as glucosides with sugar moieties, giving them higher polarity. It can be predicted that the compounds of such physicochemical properties should express a higher affinity to water fraction or butanol fraction. Generally, the more polar solvents, like methanol or aqueous methanol, are chosen for the phenolic or flavonoid extraction. The scientific works published in the previous years also practically showed the better solubility of these groups on natural compounds in the polar extraction solvent. For example, in sequential extraction method using different extraction solvents including hexane, ethyl acetate, butanol, 50% ethanol, and water, the more polar extracts are more effective in flavonoid glycosides extraction, and their solubility in ethyl acetate is rather law (https://doi.org/10.1186). Generally, it seems that the partition of between ethyl acetate is theatrically more probably to at last on the similar, like in the other works (doi: 10.4103/pr.pr_12_17), where the phenolic content in ethyl acetate and the aqueous fraction was similar. How can this discrepancy, with the theoretical polarity/affinity to water fraction and the result obtained in other studies, be explained? 2) The authors could have used the LC-MS system also to make the quantitation of the flavonoids and phenolic. This method is considered more accurate than chemical methods used in the study, which are an estimation, not a determination of the compound's content. The reagent in the Folin-Ciocalteu method does not measure only phenols but will react with any reducing substance. Also, the complexation with aluminum chloride is not specific only for flavonoids. Why wasn't the precise LC-MS method used for quantitative analysis in favor of chemical methods? 3) GC-MS results. What is the content of the compound in the fraction? Data showing the percentage of peak area is presented, but due to the specification of the GC-MS chromatography, it is not fully informative. Some compounds from the sample can be not separated on the column and, in consequence, not detected, and in MS analysis, the response of the compounds can differ due to the structural features. In the Reviewer's modest opinion, the data indicating the reflective content in the plant (presented, for example, content in 1 g of dry weight) would be more proper for the discussion. It would be interesting to compare at least the concentration of potentially active compounds (palmitic acid and sitosterol) and active concertation of the palmitic or sterol showed in cited references. 4) The GC-MS chromatogram should be included in the text or the supplementary materials. 5) Page 7. It seems that a part of the manuscript is missing. After section "GC-MS results," only table 6 is presented without discussion. 6) The UPLC-MS chromatogram can be included in supplementary material as an important result, allowing other researchers to refer to this study in the future. The Minor Essential Revisions: l. 34 and l.35 (27.97%) (16.42%) Percentage of what ? Please specify that the value indicates the peak area. l.47 considered no-economic value – a part of a phrase is missing l.57 an effective management this – some worlds are missing l. 311 What was the volume of the solvents used for the fractionation? How many times the water fraction was extracted?Author Response
Dear Respective Reviewer 1
Thank you very much for your important and valuable comments and suggestions to improve our manuscript. All of your comments and suggestions are added into the revised version, and marked by red color letters.
Please kindly find the detailed answers to each of your queries in the attached file. Your professional and academic review apparently help us to improve our manuscript.
Thank you very much
Corresponding author
Tran Dang Xuan

Reviewer 2 Report
Manuscript submitted to review deals with an investigation into biological properties - such as antioxidant, anti-α-amylase, anti-tyrosinase abilities, and cytotoxicity against CML cell lines, of Andropogon virginicus species. The overall quality of the manuscript is good however, a few issues should be addressed prior to its publication in Plants Journal.
In the Materials and methods part, the list of chemical reagents is missing.
In what solvent were the samples for antioxidant and other tests dissolved?
What was the concentration of the tested samples subjected to antioxidant tests?
What was the concentration of the control sample subjected to antioxidant tests?
What was the concentration of the DPPH radicals solution?
What was the incubation time of tested samples with ABTS and DPPH radicals before the final spectrophotometric measurement?
Were these samples kept in the dark prior to the measurements?
How did the authors find the sample volumes subjected to reactions with radicals? The too high a concentration of the tested sample will cause complete loss of the absorption band and disturb the final results.
It is hard to believe that in such a complex matrix as the plant matrix the authors identify so few compounds. To dispel these doubts, could the authors include the relevant chromatograms in Supplementary data?
Line 329: missing units of K2S2O8 concentration
Line 405: I think the first sentence in this paragraph needs correction.
In different parts of the text, Latin phrases in vivo, in silico, etc. are once written in plain text and once in italics, please unify this.
The clarification of the above ambiguities and doubts will help to raise the scientific soundness of the submitted manuscript. At this point, I suggest it be directed to major revisions.
Author Response
Dear Respective Reviewer 2
Thank you very much for your important and valuable comments and suggestions to improve our manuscript. All of your comments and suggestions are added into the revised version, and marked by red color letters.
Please kindly find the detailed answers to each of your queries in the attached file. Your professional and academic review apparently help us to improve our manuscript.
Thank you very much
Corresponding author
Tran Dang Xuan

Round 2
Reviewer 1 Report
After the first round of revision the quality of the manuscript was improved and data was described more carefully. Due to the importance of the topic, concentrationg on the alternative usage of invasive weed, as an source of antioxidant and cytotoxic compound it should be considered to publication in Plants journal.
Reviewer 2 Report
I would like to thank the authors for all the answers and corrections. After reading the revised form of the manuscript, I recommend its publication in its present form.